# Take a Breather—Physiological Correlates of a Conscious Connected Breathing Session in a Trained Group of Breast Cancer Patients

**DOI:** 10.3390/cancers17223690

**Published:** 2025-11-18

**Authors:** Alicja Heyda, Agnieszka Gdowicz-Kłosok, Magdalena Bugowska, Marcela Krzempek, Kinga Dębiec, Jolanta Mrochem-Kwarciak, Krzysztof Składowski

**Affiliations:** 11st Radiation and Clinical Oncology Department, Maria Sklodowska-Curie National Research Institute of Oncology, Gliwice Branch, 44-102 Gliwice, Poland; alicja.heyda@gliwice.nio.gov.pl (A.H.);; 2Center for Translational Research and Molecular Biology of Cancer, Maria Skłodowska-Curie National Research Institute of Oncology, Gliwice Branch, 44-101 Gliwice, Poland; 3Department of Biostatistics and Bioinformatics, Maria Sklodowska-Curie National Research Institute of Oncology, Gliwice Branch, 44-102 Gliwice, Poland; magdalena.bugowska@gliwice.nio.gov.pl (M.B.);; 4Digital Medicine Center, Maria Sklodowska-Curie National Research Institute of Oncology, Gliwice Branch, 44-102 Gliwice, Poland; 5Analytics and Clinical Biochemistry Department, Maria Sklodowska-Curie National Research Institute of Oncology, Gliwice Branch, 44-102 Gliwice, Poland

**Keywords:** breast cancer, integrative breathwork psychotherapy, conscious connected breathing, immunostimulation, cortisol, prolactin

## Abstract

This study examined the effects of a single session of Conscious Connected Breathing (CCB), which is part of Integrative Breathwork Psychotherapy (IBP), on stress levels and immune responses in breast cancer patients. Stress and negative emotions can weaken the immune system, partly due to increased cortisol levels. In contrast, prolactin may support immunity. Seventy-seven patients undergoing radiotherapy participated in a ten-session IBP program, which included rhythmic breathing through the nose and emotional expression. Blood and gas measurements were taken before and during the final session. The results showed that CCB led to mild overbreathing, with reduced carbon dioxide and oxygen levels, as well as increased blood pH. Notably, prolactin levels increased, while cortisol and IgA levels decreased. These changes suggest that breathwork may positively influence hormonal balance and immune function. This is the first study to analyze such physiological effects during CCB, indicating its potential as a complementary therapy for cancer patients. Further research is needed to assess the long-term benefits.

## 1. Introduction

High levels of stress and negative emotions have a negative impact on the treatment of malignant tumors. Effective psychosomatic interventions, such as conscious breathing techniques, reduce depression, pain, and fatigue, strengthening mental resilience and a sense of control over health [1]. A holistic approach, combining psychological and physiological aspects, promotes mutual reinforcement of these areas. Conscious breathing, irrespective of age or health status, is a technique that does not necessitate specialized equipment and has been demonstrated to possess significant therapeutic potential. A substantial body of research has demonstrated the positive impacts of rhythmic breathing on the endocrine, immune, and nervous systems.

The regulation of respiration has been demonstrated to influence interoceptive and emotional communication [2,3]. Furthermore, breathing patterns have been shown to be closely related to emotional states. As indicated by Masaoka, slow, deep breathing has been shown to synchronize with limbic system activity [4]. This synchronization modifies signals between the body and the nervous system, affecting emotions, relationships, and cognitive functions [1]. Dysfunction of the autonomic nervous system has been associated with mental disorders, and breathing techniques have been demonstrated to alleviate symptoms by influencing the vagus nerve, thereby supporting autonomic balance and stress resistance [1,5].

While slow breathing has been extensively researched, the literature on intense, deep breathing techniques—which are found in yoga, Taoism, Buddhism, and wellness practices—is less extensive [6,7]. A mounting body of research suggests that methods such as conscious connected breathing (CCB) and Sudarshan Kriya Yoga (SKY) are efficacious in enhancing emotional well-being, immune function, and reducing inflammation. CCB has been associated with improved self-esteem, mood, and sense of control [8], a decrease in depression, anxiety, and anger, and an increase in joy [9,10]. A 70% decrease in alcoholism relapses has also been reported [9], improvement in the treatment of eating disorders [11], and the occurrence of states of bliss and altered consciousness, accompanied by changes in brain waves [12].

The effects of SKY include anxiolytic, antidepressant, and analgesic properties, as well as a reduction in cortisol levels [13,14]. Additionally, it has been observed to enhance cardiac autonomic tone [15], modify brain wave activity [16], and increase NK cell count [17]. Furthermore, studies have demonstrated that SKY can enhance antioxidant status and regulate gene expression [18,19]. The extant literature on the subject is limited, and there is a paucity of studies that address the precise physiological processes occurring during a breathing session involving deep cyclic breathing. Rubin’s seminal study of two CCB sessions demonstrated altered respiratory dynamics, with divergent individual responses (hyperventilation versus CO_2_ retention), along with transient cardiovascular and neuromuscular effects [8].

A study involving 61 individuals demonstrated that active breathing techniques significantly reduce etCO_2_ levels, enhance mood, alleviate symptoms of depression, and decrease sympathetic nervous system activity. The effects persisted despite the normalization of carbon dioxide levels. Holotropic breathwork^®^ and conscious connected breathing elicited comparable physiological and psychological responses, indicating their therapeutic potential in the treatment of mental disorders [20].

A lot of studies have demonstrated the efficacy of yogic deep breathing in reducing markers of stress, oxidation, hormonal function, and immunity [18,21,22]. In an experiment involving police officers, the regular practice of Sudarshan Kriya Yoga for a period of five months resulted in decreased lactic acid levels and increased antioxidant enzymes. SOD, GSH, and catalase [18].

Chronic stress has been demonstrated to result in increased cortisol secretion, compromised immune system function, and sustained inflammation. These factors, in turn, have been shown to promote cancer development through the action of cytokines such as TNF, IL-1, and IL-6 [23,24]. Therapeutic interventions that reduce cortisol levels and stimulate prolactin have the potential to enhance well-being and bolster immunity in the battle against cancer cells.

## 2. Materials and Methods

### 2.1. Clinical Materials

The research material consisted of blood samples from patients treated with adjuvant postoperative radiation therapy due to stage I-III breast cancer at the Maria Sklodowska-Curie National Oncology Research Institute in Gliwice between 2006 and 2008. In addition, 73% of them received chemotherapy and 65% received hormonal therapy (Table 1). Participation in the study was offered to all patients admitted to the unit who met the eligibility criteria.

#### 2.1.1. Patient Eligibility Criteria for the Study

Breast cancer patients who met the following inclusion criteria were eligible for retrospective analysis:Histopathologically confirmed breast cancer.Status after previous surgical treatment.No prior treatment for another malignancy.No concurrent malignancies were diagnosed at the same time as breast cancer and during the patient’s follow-up.Patient performance status 0–1 on the ZUBROD scale.Patients over 18 years of age.

Criteria for exclusion of patients from the study:Failure to meet any of the eligibility criteria.Pregnancy.Metastatic breast cancer.Comorbidities that may interfere with the evaluation of study parameters: chronic obstructive pulmonary disease, ischemic heart disease, renal disease, alcohol and drug addiction, anorexia and bulimia, schizophrenia, Parkinson’s disease, Alzheimer’s disease, disability, dementia, advanced atherosclerosis, and organic and post-traumatic brain damage.Age > 70 years

#### 2.1.2. Rationale for the Analysis of Historical Data

The decision to conduct this analysis using data collected in 2006–2008, with subsequent analytical work performed years later, reflects several methodological and scientific considerations that warrant clarification. During the initial study period, the field of breathwork research was characterized by very limited scientific literature and isolated preliminary reports, making it challenging to establish comprehensive multidisciplinary research frameworks. The fundamental physiological principles that govern human respiratory physiology, hormonal regulation, and blood gas dynamics have remained unchanged since the original data collection, ensuring the continued validity of our measurements. Furthermore, the analytical methodologies used, including chemiluminescent immunoassays for hormone quantification and blood gas analysis techniques, represent well-established laboratory standards that have shown remarkable consistency over time, with minimal technological variations that would affect the interpretation of the results. Retrospective analysis of biological sample data is a recognized and proven method in clinical trials, especially when testing new treatments that require broad multidisciplinary knowledge for proper analysis and interpretation. The delayed publication timeline allowed for the assembly of a comprehensive research team with complementary expertise in oncology, biostatistics, molecular biology, and psychotherapy research, allowing a more thorough and nuanced analysis than would have been possible during the initial study period. This approach aligns with established practices in clinical research, where complex data sets benefit from retrospective analysis once appropriate analytical frameworks and collaborative networks have been developed.

### 2.2. Procedures

The study was carried out after receiving approval from the bioethics committee. Informed consent was obtained from all participants. Integrative Breathwork Psychotherapy (IBP) consisted of 10 sessions, three times a week. The Conscious Connected Breathing session lasted 45 min, followed by 15 min of free emotional expression. On other days, patients practiced CCB for 15 min with the psychotherapist or by themselves alone. The classes were held in small groups (up to six people); see Figure 1. The patients did not receive other psychotherapy or counseling during the study period.

The structured protocol of ten prolonged breathwork sessions was developed to ensure that the participants could fully benefit on a psychosomatic level and properly master the breathing technique. This approach aligns with existing research frameworks, where a series of sessions is necessary to achieve effective and sustained results. A similar protocol was used in the studies conducted by Sharma [18], who implemented repeated training sessions of Sudarshan Kriya Yoga, highlighting the importance of systematic practice to induce measurable physiological changes. Furthermore, this session structure resonates with the original standardized guidelines for Conscious Connected Breathing developed in the 1970s, as documented by Orr and Ray (1983) [25] and Orr (1988) [26]. These foundational works emphasized the need for multiple sessions to help participants attain technical proficiency and navigate the associated psychosomatic experiences effectively. By conducting measurements during the tenth session, the study ensured that the participants had reached a level of competence and integration that allowed a valid assessment of the physiological effects of the intervention.

IBP combines intensive breathwork training and the state of mindfulness, followed by free emotional expression. The breathing work session is followed by an expression based on a verbal process. Patients communicate their feelings and give them their own meaning. Patients are supported to maintain non-judgmental mindfulness during any kind of experience while breathing.

Conscious Connected Breathing (CCB) includes taking moderately deep breaths, deeper than usual; one after another, without pausing, rhythmically drawing air into the upper lungs. The exhalation is completely relaxed, and the air leaves the lungs without any effort: pushing, dividing the exhalation into parts, or holding back. Breathing was carried out through the nasal respiratory passage. Before starting IBP, patients received information on the experience of the specifics of the conscious connected breathing session, including possible temporary tingling, slight sleepiness, and emotional experiences.

Integrative Breathwork Psychotherapy Session: After approximately 3 min of calming the breath and relaxing the muscles, the practitioner begins breathing rhythmically in the CCB described above. Proper breathing during subsequent breathwork sessions was guided by an experienced breath worker and certified psychotherapist. The therapist helped reach an elemental state of mindfulness during the breathwork session, focus, and allowed emotions and thoughts to flow freely without attaching or pushing them away. No specific suggestions were made about the origin of session sensations. The patients received emotional support to allow them to open up to new experiences and feel safe.

### 2.3. Data Collection

The patients in the experimental group participated in an intensive cycle of IBP as part of a larger study. The first measurement was taken at rest before the start of session no. 10.

All blood samples were taken at the same hour due to fluctuations in diurnal cortisol. The second measurement was taken at the 30th minute of the breathing session. This measurement time is based on rhythmic breathing in Sudarshan Kriya Yoga of Sharma et al. [18], where the trained subjects achieved significant biological effects after 30 min of circular breathing. It showed that this time of gathering blood samples within the breathwork session is suitable for seeing significant hormonal fluctuations. Blood collected was transferred to the laboratory immediately after sampling.

### 2.4. Reasons for Selecting the Above Parameters

#### 2.4.1. Arterialized Capillary Blood Gas Analysis

The selected parameters enable a multidimensional evaluation of respiratory therapy effectiveness: pO_2_, sO_2_, ctO_2_, O_2_Hb, and HHb.

Assessment of blood oxygenation: pCO_2_, pH, HCO_3_^−^, BE, and BB for assessing acid-base balance and ventilation.

COHb and MetHb monitor abnormal hemoglobin forms.

These parameters were selected to monitor the effectiveness and safety of therapy, i.e., to evaluate whether it improves oxygenation and ventilation or causes undesirable changes in hemoglobin.

#### 2.4.2. Hormones: Cortisol and Prolactin

Cortisol may increase in response to hypoxia, respiratory stress, or medical interventions. Its level may correlate with the severity of the clinical condition and the neuroendocrine response to therapy.

Prolactin: Its concentration may be modulated by physiological stress, drowsiness, and hypoxia. It may serve as an indicator of the body’s response to improved respiratory comfort.

These hormones can support the assessment of the systemic response to therapy, especially in the context of stress and adaptation.

#### 2.4.3. Immunoglobulin A (IgA)

As a marker of mucosal immunity, IgA may be relevant in the context of chronic lung disease, infection, or airway inflammation. Its level may change in response to improved ventilation and reduced inflammation.

IgA measurement can support assessing the immune component of the response to respiratory therapy, particularly for chronic conditions or interventions that promote airway clearance.

Physiological measurements were intentionally conducted exclusively at the tenth session after participants had achieved technical proficiency through repeated training, rather than at earlier time points. This design enabled us to characterize the acute within-session physiological response profile of competent CCB practice, unconfounded by learning effects or technical inconsistency. Longitudinal assessment of cumulative training-induced adaptations in baseline neuroendocrine and immune parameters across the full intervention period (baseline through 12-week follow-up) will be reported separately.

### 2.5. Analytical Methods

CBC analysis was performed using a Sysmex XN-2000 hematology analyzer (Sysmex America, Inc., Lincolnshire, IL, USA). Sysmex XN-2000 is a multi-parameter blood cell counter. To count blood cells, the auto-machine uses an impedance principle. The impedance principle uses a constant electric current that is passed through a blood sample and a reagent solution to determine the changes in electrical resistance that occur when blood cells pass through the detection aperture. To maintain the quality of the CBC analyzer in the known blood samples (normal, low, and high), background checks and maintenance of the machine were performed according to the manufacturer’s instructions and according to the Clinical Laboratory Institute. This study aimed to qualify blood gas parameters (pH, PCO_2_, pO_2_), electrolytes (Na^+^, K^+^), ionized calcium (Ca^2+^), metabolites (glucose, lactate), and oximetry parameters (tHb, O_2_Hb, COHb, MetH). Using the Cobas b221 analyzer, pH, pCO_2_, Na^+^, K^+^, and Ca^2+^ are measured using potentiometry methods, pO_2_, glucose, and lactate using the amperometry method. tHb, O_2_Hb, COHb, and MetHb are measured by spectrophotometry in capillary blood.

The concentration of IgA, prolactin, and cortisol was determined in blood serum. Blood was obtained under standard conditions, with patients in a fasting state, between 7.00 a.m. and 9.00 a.m., using a vacuum Becton–Dickinson system, into sample tubes without anticoagulant. Samples obtained after centrifugation at 3000 rotations/min for 10 min. At 40 °C were analyzed on the same day. The IgA concentration was determined by a nephelometric immunoassay, using the Atelica analyzer and the commercial kit analyzer from Siemens Healthcare (Tarrytown, NY, USA). The concentration of cortisol was determined by chemiluminescent immunoassay (CLIA) using Siemens Healthcare and Immulite 2000i reagent kits (Tarrytown, NY, USA). The concentration of prolactin was determined by a chemiluminescent microparticle immunoassay (CMIA), using an Atelica analyzer (Siemens Healthcare Diagnostics Inc., Tarrytown, NY, USA) and a commercial kit analyzer from Abbott Laboratories (Abbott Park, IL, USA).

### 2.6. Statistical Analysis

Continuous variables are presented as means and standard deviations, with normality assessed using the Shapiro–Wilk test. Before analysis, data were log-transformed to approximate normal distribution and stabilize variance, except for variables already expressed as logarithms (e.g., pH) and percentages. To compare pre- and post-IBP session measurements, paired *t*-tests were performed. The magnitude of the differences between the time points was quantified using Cohen’s d as the effect size. Reported *p* values are unadjusted due to the small sample size (*n* = 48), which limits the utility of multiple comparison corrections and helps avoid inflating the risk of Type II errors. Given the retrospective design and the fixed sample size, the minimum detectable effect (MDE) was calculated with 80% statistical power (α = 0.05) to support the interpretation of both significant and nonsignificant results. All statistical analyses were performed using R software (version 4.4.1; R Foundation for Statistical Computing, Vienna, Austria). A two-sided *p* < 0.05 was considered statistically significant.

## 3. Results

### 3.1. Participant Characteristics

A total of 93 patients were enrolled in the study. Fifty-six patients (60%) agreed to participate. During the experiment, eight patients were excluded from the analysis for the following reasons: metastases detected during treatment; withdrawal of consent.

A total of 48 patients were included in the final analysis. The demographic and clinical characteristics of the study population are summarized in Table 2.

### 3.2. Changes in Hormonal, Immune, and Gasometric Arameters

Table 3 presents the results of the parameters evaluated, grouped into two categories: hormonal and immune markers (cortisol, prolactin, and IgA), and gasometric parameters (pH, pCO_2_, pO_2_, base excess, and other oxygen-related variables). The box plots in Figure 2 illustrate the changes in these parameters, highlighting individual data points before and after the Conscious Connected Breathing session, along with the direction of change for each patient.

### 3.3. Hormonal and Immune Markers

All three markers tested showed a significant change during the IBP session, with a moderate effect size (Table 3). The cortisol level was reduced (*p* < 0.001; *d* = −0.59), whereas prolactin level demonstrated a notable increase (*p* < 0.001; *d* = 0.54). Furthermore, a decrease in IgA level was also observed (*p* < 0.001; *d* = 0.56). These findings are presented in Figure 2A.

### 3.4. Gasometry

Subsequent to the completion of the IBP session, substantial statistically significant alterations were detected in multiple blood gas parameters. An increase in blood pH (*p* < 0.001; *d* = 0.64) and base excess (BE, *p* < 0.001; *d* = 0.51). At the same time, a substantial decline was observed in the partial pressure of oxygen (pO_2_, *p* < 0.001; *d* = −0.57) and the partial pressure of carbon dioxide (pCO_2_, *p* = 0.003; *d* = −0.45), hydrogen ion concentration (cH^+^, *p* < 0.001; *d* = −0.66), and the total oxygen content (ctO_2_, *p* < 0.001; *d* = −0.60).

Among hemoglobin parameters, a decrease was observed in total hemoglobin content (ctHb, *p* = 0.001; *d* = −0.50) and hemoglobin oxygen saturation (O_2_Hb, *p* = 0.011; *d* = −0.38), in contrast to deoxygenated hemoglobin concentration (HHb, *p* = 0.004; *d* = 0.44), where an increase was recorded.

For BB, sO_2_, cHCO_3_, and MetHb, no significant changes were detected, and the obtained effect sizes were below the minimum detectable effect (|*d|* < 0.41).

## 4. Discussion

This study corroborates the therapeutic potential of conscious connected breathing (CCB) as a complementary modality in oncology. CCB exerts a regulatory influence over the nervous, endocrine, and immune systems, thereby enhancing patient well-being and potentially improving treatment outcomes.

Chronic stress, a common occurrence among cancer patients, activates the sympathetic nervous system and the hypothalamic–pituitary–adrenal (HPA) axis, resulting in dysregulated cortisol and catecholamine secretion. These stress hormones modulate immune responses and promote tumor progression [27,28,29]. Behavioral disturbances, including anxiety and depression, have been associated with systemic immunosuppression, as indicated by reduced cytokine production and diminished T cell proliferation [30,31,32].

The present findings demonstrate that CCB significantly reduces cortisol levels, indicating a reduction in stress, and increases prolactin (PRL), a hormone with immunostimulatory properties. As Borba et al. (2018) [33] and Alemán-Garcia et al. (2021) [34] have demonstrated, PRL has been shown to promote T cell activation and B cell maturation, thereby enhancing immunoglobulin production. This hormonal shift may counteract the immunosuppressive effects of glucocorticoids and support immune resilience [33,34].

The results of the blood gas analysis indicated elevated pH, increased base excess (BE), and elevated deoxygenated hemoglobin (HHb), along with decreased pCO_2_ and hydrogen ion concentration (cH^+^), suggesting an enhanced acid-base balance. The acidic nature of the tumor microenvironment (a phenomenon known as the Warburg effect) suggests that such shifts may impede cancer cell proliferation and enhance therapeutic responsiveness. The stability observed in parameters such as sO_2_, cHCO_3_, and MetHb suggests that respiratory intervention did not disrupt the body’s internal equilibrium [35].

A multitude of studies have demonstrated that Holotropic Breathwork^®^ and CCB result in a reduction in end-tidal CO_2_ (etCO_2_), thereby inducing altered states of consciousness and enhancing mood and depressive symptoms [20]. Conversely, neofunctional deep breathing (NDB) has been shown to reduce allostatic load and inflammatory markers in response to psychosocial stress, thereby validating its potential as a noninvasive intervention targeting HPA axis dysregulation [36].

In light of the World Health Organization’s (WHO) data identifying stress as a major contributor to “diseases of civilization” [37], accessible interventions such as breathwork are imperative. Regular practice has been demonstrated to enhance natural killer (NK) cell activity [17,22], modulate the tumor microenvironment, and support immune function.

In summary, CCB shows potential as a noninvasive adjunct therapy, capable of modulating stress-related hormonal and immune parameters and influencing the tumor microenvironment. Further research is warranted to elucidate its mechanisms and optimize clinical protocols.

## 5. Limitations

The present study has several important limitations. To begin with, the sample size was relatively small, which may reduce the statistical power and generalizability of the findings. Moreover, the study group consisted exclusively of clinical patients rather than a broader or healthier population, which can limit the applicability of the results to other contexts. In addition, emotional states were not evaluated before and after the session; this decision was made because invasive blood sampling immediately prior to questionnaire administration could have influenced actual emotional status and distorted the measurement. Furthermore, respiratory parameters such as breathing frequency, depth, and heart rate were not monitored during the intervention, preventing the analysis of possible physiological mechanisms underlying the observed changes. Finally, the single session lacked both a comparison group of healthy women and a control group that underwent relaxation rather than breathwork, limiting the interpretation of the effects as specific to the intervention. Given a fixed sample size (*n* = 48), the minimum detectable effect size (Cohen’s *d)* was 0.41 for 80% statistical power at α = 0.05. This means that only medium-to-large effects could be reliably detected. Smaller effects, even if present, may not reach statistical significance. Therefore, interpretation of the data focuses not only on *p* values but also on the magnitude of observed effects and their potential biological relevance.

## 6. Conclusions

This study is a pioneering analysis of changes in capillary blood gas levels and hormonal changes during Conscious Connected Breathing sessions. This study provides new information, demonstrating changes in hormonal balance, suggesting possible modulation of hormonal and immunological markers through deep breathing therapy. The results obtained suggest that CCB sessions have a significant impact on acid-base balance and blood gas exchange, which may indicate adaptive changes in body physiology in response to the intervention used. Interpreting these changes necessitates further research to determine their potential mechanisms and consequences for bodily function. Significant positive changes in the functioning of the endocrine, immune, and respiratory-circulatory systems that our group observed in such a brief period during breathwork suggest that this type of therapy has a substantial impact when used as a complementary therapy in cancer patients Figure 3. To further elucidate the therapeutic mechanisms underlying these observations, additional research is necessary, including enrollment of larger patient cohorts and analysis of the long-term impact of breathing therapy on psychological and immunological responses.

## Figures and Tables

**Figure 1 cancers-17-03690-f001:**
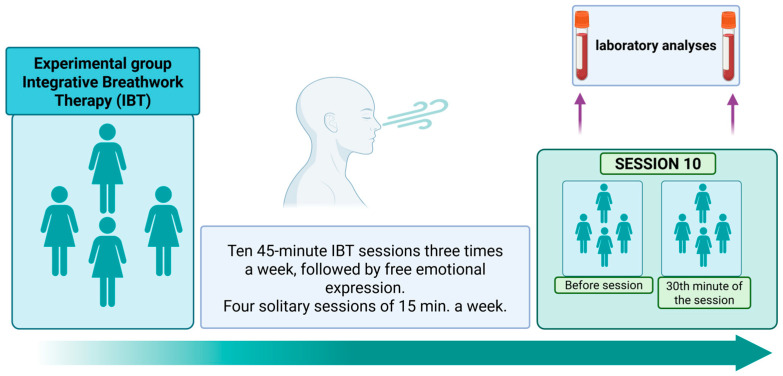
Diagram of the IBP procedure in the experimental group. Own illustration created with BioRender (https://BioRender.com/ui4uct, accessed on 1 October 2025).

**Figure 2 cancers-17-03690-f002:**
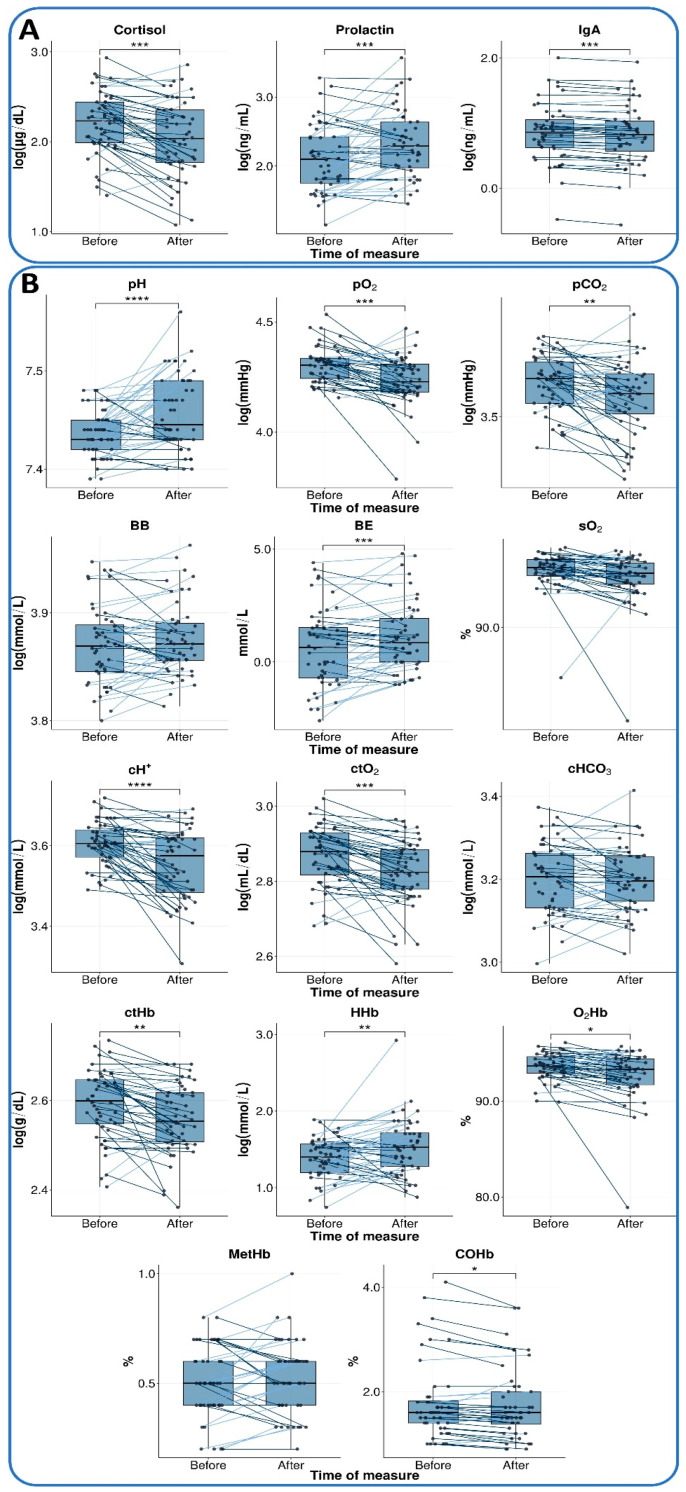
Changes in parameters before and after IBP. The box plots illustrate changes between the measure at rest 15 min before session 10 (measure I, before) and 30 min into the session (measure II, after) for two main categories: (**A**) hormonal and immune markers and (**B**) gasometric parameters. The median is represented by a horizontal line within each box, and individual observations are connected by lines to indicate the direction of change. Observations with an increased value are connected by light blue lines, whereas those with a decrease are connected by dark blue lines. Pre–post differences were analyzed using a paired *t*-test (see Statistical Analysis for details). Statistically significant differences are marked with an asterisk in the following convention for symbols: *—*p* <= 0.05, **—*p* <= 0.01, ***—*p* <= 0.001, ****—*p* <= 0.0001.

**Figure 3 cancers-17-03690-f003:**
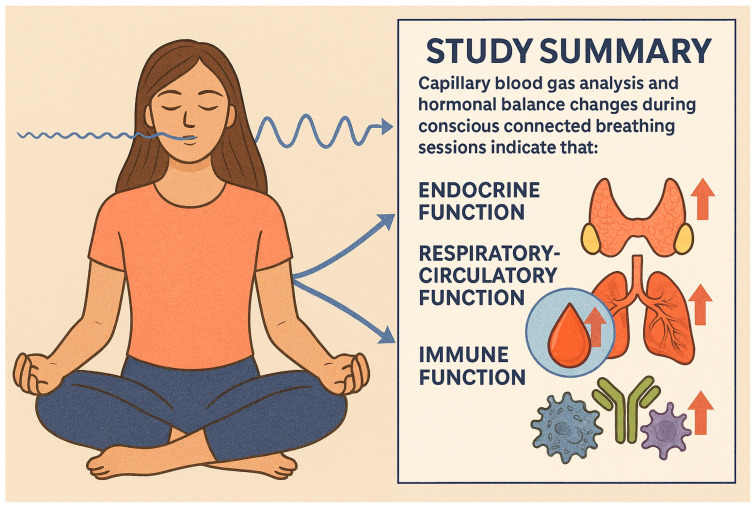
Graphical summary of the study.

**Table 1 cancers-17-03690-t001:** Measurements of biological variables in the experimental group.

Measurement No. (Time)	Physiological Variables
I (at rest, 15 min before session no. 10)	Arterialized capillary blood gasometry: pO_2_, pCO_2_, pH, cH^+^, cHCO_3_, BE, BB, sO_2_, ctO_2_, MetHb, COHb, O_2_Hb, HHb Cortisol and prolactin concentrations IgA
II (at 30 min in session no. 10)	Same as above

**Table 2 cancers-17-03690-t002:** Baseline characteristics of the study group.

Characteristic	Patients (*n* = 48)
Age—mean (range)	53 (38–68)
Education *n* (%)	
Primary	9 (19)
Vocational	9 (19)
Secondary	19 (39)
Higher	11 (23)
Social status *n* (%)	
Married	34 (71)
Divorced	1 (2)
Widowed	9 (19)
Single	4 (8)
Smoking status *n* (%)	
Yes	9 (19)
No	39 (81)
Type of treatment	
Surgical treatment *n* (%)	
Mastectomy	25 (52)
Conservative Surgery	23 (48)
Chemiotherapy *n* (%)	
Adjuvant	19 (40)
Neoadjuvant	16 (33)
Hormonal therapy *n* (%)	31 (65)
Area of radiotherapy *n* (%)	
Breast or chest wall with scar	25 (52)
Breast/chest + axilla	3 (6)
Breast/chest + axilla + supraclavicular	20 (42)
Radiation dose *n* (%)	
45 Gy/18 fx	9 (19)
50 Gy/25 fx	39 (81)
Additional boost to the surgical bed (10 Gy)	15 (31)
Receptor status	
ER status *n* (%)	
Negative	15 (31)
1 +	9 (19)
2 ++	14 (29)
3 +++	9 (19)
No data	1 (2)
PR status *n* (%)	
Negative	17 (35)
1 +	9 (19)
2 ++	11 (23)
3 +++	9 (19)
No data	2 (4)
HER status *n* (%)	
Negative	18 (38)
1 +	14 (29)
2 ++	4 (8)
3 +++	3 (6)
No data	9 (19)

Abbreviations: ER—Estrogen receptor, PR—Progesterone receptor, HER—Human epidermal growth factor receptor 2 (HER2). The designation +++; ++; +—is the level of receptor expression, which indicates the hormone dependence of the tumor.

**Table 3 cancers-17-03690-t003:** Characteristics of parameters before and after IBP intervention.

	Measure I (*n* = 48)	Measure II (*n* = 48)				
	Mean ± SD	Mean ± SD	*p* Value	Effect Size (95% CI)	Effect Size Interpretation	Change Direction
Hormonal and Immune Markers						
Cortisol	9.6 ± 3.2	8.4 ± 3.5	<0.001	−0.59 (−1.01 to −0.29)	medium	↓
Prolactin	9.3 ± 5.1	11.5 ± 6.4	<0.001	0.54 (0.33 to 0.76)	medium	↑
IgA	2.6 ± 1.2	2.5 ± 1.2	<0.001	−0.56 (−0.88 to −0.32)	medium	↓
Gasometry						
pH	7.4 ± 0.0	7.5 ± 0.0	<0.001	0.64 (0.39 to 0.91)	medium	↑
pO_2_	74.1 ± 6.2	69.4 ± 7.6	<0.001	−0.57 (−0.89 to −0.31)	medium	↓
pCO_2_	37.7 ± 3.8	35.9 ± 4.6	0.003	−0.45 (−0.72 to −0.19)	small	↓
BB	48.0 ± 1.8	48.2 ± 1.6	0.06	0.28 (−0.03 to 0.6)	small	↔
BE	0.6 ± 1.7	1.1 ± 1.5	<0.001	0.51 (0.25 to 0.84)	medium	↑
sO_2_	95.6 ± 1.9	94.9 ± 2.5	0.09	−0.25 (−0.64 to −0.01)	small	↔
cH^+^	36.9 ± 2.0	35.0 ± 3.0	<0.001	−0.66 (−0.94 to −0.42)	medium	↓
ctO_2_	17.7 ± 1.4	16.9 ± 1.4	<0.001	−0.60 (−0.95 to −0.31)	medium	↓
cHCO_3_	24.6 ± 2.1	24.6 ± 1.9	0.98	0.00 (−0.28 to 0.30)	negligible	↔
ctHb	13.4 ± 1.0	12.9 ± 1.0	0.001	−0.50 (−0.85 to −0.24)	medium	↓
HHb	4.1 ± 1.0	5.0 ± 2.5	0.004	0.44 (0.16 to 0.80)	small	↑
O_2_Hb	93.6 ± 1.4	92.7 ± 2.7	0.01	−0.38 (−0.75 to −0.19)	small	↓
MetHb	0.5 ± 0.2	0.5 ± 0.2	0.20	0.19 (−0.10 to 0.50)	small	↔
COHb	1.8 ± 0.7	1.7 ± 0.7	0.03	−0.32 (−0.68 to −0.03)	small	↓

Interpretation of the effect size in accordance with the rules of thumb on the magnitudes of effect sizes. Retrieved 12 March 2025, from https://imaging.mrc-cbu.cam.ac.uk/statswiki/FAQ/effectSize (accessed on 13 November 2025). Abbreviations: SD, standard deviation; CI, confidence interval. ↑ increase in level; ↓ decrease in level; ↔ no change.

## Data Availability

All data (anonymized) are available from the authors and can be shared with anyone who is interested. To access the data, send your request to Alicja Heyda.

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
