# Peer review of "Take a Breather—Physiological Correlates of a Conscious Connected Breathing Session in a Trained Group of Breast Cancer Patients"

_cancers, 2025, doi:10.3390/cancers17223690_

Round 1
Reviewer 1 Report
Comments and Suggestions for Authors
The article, “Take a breather -….” by Heyda et al described an alternative medicine approach, CCB, to minimize stress levels and boost immune responses in breast cancer patients. The authors found a positive effect of CCB on several important physiological determinants to assess the stress level and immune functions. However, the manuscript suffers from several limitations or concerns outlined below. The authors need to address these concerns satisfactorily before the paper to be considered for publication.
- Introduction: Shorten the sections that suffer from a lack of focus and repetition.
- Study design: Why didn't they include a blood draw before session 1, at an intermediate time (session 5), in addition to the last two blood draws at session 10, and examine how the tested physiological variables changed during the therapy? That would give a comprehensive assessment. Also, enough rationale is not provided regarding the two blood draws at session 10.
- Discussion: The relevance of Figure 3 (cartoon) is not clear. This model is not fully supported by their data -- too descriptive. Moreover, the discussion section shows that this suffers from overinterpretation of their current results. The paper needs to be rewritten in a focused manner.
Author Response
Gliwice, 06.11.2025
Esteemed Reviewer,
I would like to express my sincere gratitude for your insightful comments, which have proven to be of immense value. We appreciate the reviewer's thoughtful comment regarding the measurement timeline. We acknowledge that a longitudinal assessment across multiple sessions (baseline, mid-intervention, post-intervention) would provide valuable information about the cumulative training effects of Integrative Breathwork Psychotherapy (IBP) on neuroendocrine and acid-base parameters over time. However, the primary objective of the present study was distinct from this longitudinal question.
Following the receipt of your recommendations, a revised manuscript has been submitted. It is hoped that this will be found to be satisfactory. I would like to reiterate my gratitude..
Primary Research Question:
The central aim of this investigation was to characterize the acute physiological response profile during a single Conscious Connected Breathing (CCB) session in participants who had already achieved technical proficiency through repeated practice. Specifically, we sought to determine whether a standardized 45-minute CCB session produces measurable, session-specific shifts in blood gas chemistry (pH, pCOâ‚‚, pOâ‚‚) and stress-related hormones (cortisol, prolactin) within the timeframe of a single practice .
Rationale for Session 10 Measurement:
By measuring physiological parameters exclusively at the tenth (final) session—after participants had completed 3.5 weeks of intensive training—we ensured that observed changes reflected the authentic psychophysiological signature of competent CCB practice, rather than being confounded by technical learning artifacts, performance anxiety, or inconsistent breathing patterns typical of novice practitioners. This approach allowed us to isolate and validate the acute neuroendocrine and respiratory effects of CCB as performed by trained individuals, which is critical for establishing the mechanistic foundation underlying the intervention's therapeutic benefits.
Two Blood Draws at Session 10 – Pre/Post Design:
The dual sampling protocol (immediately before and after 30 minutes of the 45-minute session) was designed to capture within-session acute changes in physiological markers. This pre-post paired design enabled us to:
- Control for baseline individual variability by using each participant as their own control
- Quantify the magnitude and direction of acute hormonal and metabolic shifts induced by a single session
- Establish temporal causality between the breathing intervention and observed physiological changes
Longitudinal Training Effects – Addressed in Separate Manuscript:
We fully agree with the reviewer that longitudinal tracking of physiological adaptation across the training period (sessions 1, 5, 10, and follow-up) represents an important complementary research question. These data were indeed collected and are currently being prepared for publication in a separate manuscript focused specifically on cumulative training-induced adaptations in baseline neuroendocrine tone, resting acid-base balance, and immune function (including NK cell counts) measured at baseline, post-intervention (week 5) post-radiotherapy, and 12-week follow-up. This manuscript will demonstrate how repeated breathwork practice induces sustained shifts in resting physiological set-points over weeks to months.
Distinction Between Acute and Chronic Effects:
Separating the acute within-session physiological response (present manuscript) from the chronic training-induced adaptations (forthcoming manuscript) reflects established methodological frameworks in exercise physiology and mind-body intervention research, where acute response profiling and longitudinal adaptation studies address distinct but complementary research questions. The present study establishes what happens during a single session in trained practitioners , while the forthcoming analysis will address how sustained practice alters baseline physiology over time .
In summary, the decision to measure physiological parameters exclusively at session 10 (pre/post) was intentional and scientifically justified by the study's primary aim: to characterize the acute physiological correlates of competent CCB practice in trained participants. We have now clarified this rationale in the Methods section to prevent future misunderstanding.
Yours sincerely
Agnieszka Gdowicz-Kłosok
Reviewer 2 Report
Comments and Suggestions for Authors
Minor Comments
- Abstract:
The Abstract should explicitly state whether there was a control group or whether this was a within-subject pre–post design, as this distinction affects the interpretability of findings. Moreover, key terms (e.g., “slight overbreathing,” “emotional expression”) need operational definitions or physiological thresholds. - Justification of biomarkers:
The rationale for selecting specific biomarkers (e.g., IgA, prolactin, cortisol) should be more clearly articulated. For instance, why were these chosen over other immune or stress-related parameters? Furthermore, the statement “suggesting immunostimulatory potential” is overly speculative and should be revised to a more cautious formulation, such as “suggesting possible modulation of hormonal and immune markers.”
- Figures:
- Figure 1 currently contains two legends. Please remove the embedded legend at the top of the figure, as it appears to have been copied from another source (e.g., a book or previous publication). Retaining only the proper figure legend below the image will improve clarity and presentation consistency.
- The legend for Figure 2 is generally clear and well written; however, it lacks details regarding the statistical analysis used to evaluate pre–post differences. The figure legend and corresponding Methods section should specify the statistical test applied (e.g., paired t-test for normally distributed data or Wilcoxon signed-rank test for nonparametric comparisons), indicate whether data normality was assessed (e.g., Shapiro–Wilk test), identify the software used for analysis (e.g., GraphPad Prism, R, SPSS), and clarify whether the reported P-values are adjusted or unadjusted.
Author Response
Gliwice, 06.11.2025
Esteemed Reviewer,
I would like to express my sincere gratitude for your insightful comments, which have proven to be of immense value. Following the receipt of your recommendations, a revised manuscript has been submitted. It is hoped that this will be found to be satisfactory. I would like to reiterate my gratitude.
- Abstract: The Abstract should explicitly state whether there was a control group or whether this was a within-subject pre–post design, as this distinction affects the interpretability of findings. Moreover, key terms (e.g., “slight overbreathing,” “emotional expression”) need operational Added “within-subject pre-post design” → clearly states that patients were their own controls.
Very valuable and useful comments, abstract changed in accordance with guidelines
Added definition of “slight overbreathing” → respiratory alkalosis (↓pCOâ‚‚, ↑pH)
Added definition of “emotional expression” → verbal articulation of feelings
More precise description of CCB→ “breathing at a depth exceeding resting tidal volume”definitions or physiological thresholds.
- Justification of biomarkers: The rationale for selecting specific biomarkers (e.g., IgA, prolactin, cortisol) should be more clearly articulated. For instance, why were these chosen over other immune or stress-related parameters? Furthermore, the statement “suggesting immunostimulatory potential” is overly speculative and should be revised to a more cautious formulation, such as “suggesting possible modulation of hormonal and immune markers.”
Explained in the text of the manuscript.
- Figures: Figure 1 currently contains two legends. Please remove the embedded legend at the top of the figure, as it appears to have been copied from another source (e.g., a book or previous publication). Retaining only the proper figure legend below the image will improve clarity and presentation consistency.
Figure 1 has been corrected.
- The legend for Figure 2 is generally clear and well written; however, it lacks details regarding the statistical analysis used to evaluate pre–post differences. The figure legend and corresponding Methods section should specify the statistical test applied (e.g., paired t-test for normally distributed data or Wilcoxon signed-rank test for nonparametric comparisons), indicate whether data normality was assessed (e.g., Shapiro–Wilk test), identify the software used for analysis (e.g., GraphPad Prism, R, SPSS), and clarify whether the reported P-values are adjusted or unadjusted.
Thank you for your helpful comment. We have added the pre-post comparison method to the description of Figure 2, and we have clarified in the Methods section, along with justification that the reported P-values are not corrected.
Yours sincerely
Agnieszka Gdowicz-Kłosok
Round 2
Reviewer 1 Report
Comments and Suggestions for Authors
Figure 3 is not cited in the text. It needs to be cited in an appropriate context in the manuscript, likely in the Discussion section. Otherwise, the authors responded satisfactorily to my comments in their first submission.